# ICT Framework for Supporting Applied Behavior Analysis in the Social Inclusion of Children with Neurodevelopmental Disorders

**DOI:** 10.3390/s23156914

**Published:** 2023-08-03

**Authors:** Sara Jayousi, Alessio Martinelli, Paolo Lucattini, Lorenzo Mucchi

**Affiliations:** 1Department of Information Engineering, Polo Universitario “Città di Prato”, 59100 Prato, Italy; 2Telespazio (Leonardo & Thales Company), 00156 Rome, Italy; alessio.martinelli@telespazio.com; 3Department of Movement, Human and Health Sciences, University of Rome “Foro Italico”, 00135 Rome, Italy; paolo.lucattini@uniroma4.it; 4Department of Information Engineering, University of Florence, 50139 Florence, Italy; lorenzo.mucchi@unifi.it

**Keywords:** Information and communication technologies, ICT, wireless communications, applied behavior analysis, ABA, children with neurodevelpmental disorders, social inclusion

## Abstract

The applied behavior analysis (ABA) model emphasizes observable and measurable behaviors by carrying out decision making using experimental data (behavioral observation assessment strategies). In this framework, information and communication technology (ICT) becomes highly suitable for enhancing the efficiency and effectiveness of the methodology. This paper aims to delve into the potential of ICT in providing innovative solutions to support ABA applications. It focuses on how ICT can contribute to fostering social inclusion with respect to children with neurodevelopmental disorders. ICT offers advanced solutions for continuous and context-aware monitoring, as well as automatic real-time behavior assessments. Wireless sensor systems (wearable perceptual, biomedical, motion, location, and environmental sensors) facilitate real-time behavioral monitoring in various contexts, enabling the collection of behavior-related data that may not be readily evident in traditional observational studies. Moreover, the incorporation of artificial intelligence algorithms that are appropriately trained can further assist therapists throughout the different phases of ABA therapy. These algorithms can provide intervention guidelines and deliver an automatic behavioral analysis that is personalized to the child’s unique profile. By leveraging the power of ICT, ABA practitioners can benefit from cutting-edge technological advancements to optimize their therapeutic interventions and outcomes for children with neurodevelopmental disorders, ultimately contributing to their social inclusion and overall wellbeing.

## 1. Introduction

In a time when technology has a crucial impact on people’s lives, the benefits can be much more evident when the technology is applied in supporting people with disabilities. Starting from the results achieved in [1], which described the potentialities of information and communication technology (ICT) in creating opportunities for the social inclusion of children with disabilities via playful and entertaining activities, the objective of this study is to analyze the role of ICT in the framework of educative and rehabilitative interventions connected to the implementation of applied behavior analysis (ABA). Starting from the results achieved in [1], which described the potentialities of ICT in creating opportunities for the social inclusion of children with disabilities via playful and entertaining activities, the objective of this study is to analyze the role of ICT in the framework of educative and rehabilitative interventions connected to the implementation of ABA. Specifically, methods by which the ICT can support the methodological application of ABA therapy in children with neurodevelopmental disorders are presented.

It is worth highlighting that with the term “ICT”, we refer to all technologies that enable the transmission, reception, and processing of data and information. The term primarily focuses on communication technologies (such as the Internet, wireless communication networks and systems, etc.) and not only includes all mechanisms that can access information (communication access technologies, protocols, and interfaces) but also digital technological solutions. In contrast, with the term “support”, we refer to the efficacy and sustainability of interventions. In the present context, the interventions refer to ABA therapy for children with neurodevelopmental disorders.

ABA can be defined as the science in which the principles of behavioral analyses are systematically applied in order to improve personal and social abilities, and observational assessments are carried out to determine the variables that are responsible for behavioral changes.

ICT may play a crucial role in making available the necessary technological support for observational assessments. ICT can provide advanced solutions for defining a continuous and context-aware monitoring system that can be used for a remote and real-time assessment of behavior.

The objective of the research is to analyze how ICT can be utilized in order to provide innovative solutions for supporting the application of ABA. Specifically, the study focuses on promoting social inclusion for children with neurodevelopmental disorders. By examining the potential of ICT in enhancing the efficiency and effectiveness of the ABA methodology and functional behavioral assessments, this research study aims to identify how ICT can contribute to improving outcomes for children with neurodevelopmental disorders and how ICT can foster their social inclusion. Additionally, this research study aims to explore the role of wireless sensor systems and artificial intelligence algorithms in real-time behavioral monitoring, intervention guidance, and personalized behavior analysis within the context of ABA therapy. Details on this research study’s workflow are provided in the following subsection.

### Aim of the Paper

Addressing the need of conceptualizing, designing, and creating products and services with an inclusive perspective, the main objective of this interdisciplinary work is to study how ICT can be exploited to improve the efficiency and effectiveness of ABA application methodologies and create inclusive opportunities for children with neurodevelopmental disorders by reducing barriers and increasing facilitators.

This paper aims to analyze how ICT can provide innovative solutions to support the application of the ABA model, with a particular focus on fostering social inclusion for children with neurodevelopmental disorders.

Therefore, as depicted in Figure 1, the problem statement can be synthesized using the following question: *What are the main issues of social inclusion for children with neurodevelopmental disorders and how ICT can be used to cope with them?* Starting from the review of the state of the art on social inclusion for children with neurodevelopmental disorders [1], the adoption of the ABA model for improving children’s behavior is analyzed. The features of the ABA model and the main methodological needs for easing its application are identified to guide the study of ICT’s role in such a context.

In particular, an analysis of the existing link between ABA and ICT is carried out to identify the main current shortcomings and to investigate how the adoption of new available technologies can effectively support the ABA method via their advanced potentialities. This research study’s workflow (Figure 1) allows the definition of an innovative ICT service platform that is specifically designed for the application of the ABA model. This represents the main result presented in this paper. It is worth highlighting that the design of the platform comes from an interdisciplinary investigation that relies on the approach of a participatory design to satisfy the needs of the end-user (e.g., ABA therapists) and improve their quality of experience. This enables the definition of a real use case that was taken as a reference to show the benefit of the proposed system with respect to already existing and sometimes adopted technologies for the application of ABA therapy. Moreover it not only allows the identification of the main potential challenges that need to be taken into account both in the development and validation phases but also allows the system’s acceptance and, therefore, its adoption by the end-users.

The paper is organized as follows: The application context of the proposed study is introduced in Section 2 by providing an overview of the concepts of neurodevelopmental disorders and social inclusion. Section 3 reports the main features of the ABA model, highlighting both the main benefits for social inclusion and shortcomings in terms of methodological aspects; Section 4 focuses on the analysis of the role of ICT in providing solutions for supporting the application of the ABA model, highlighting both the existing ICT-ABA link and the new potentialities of innovative technologies for the enhancements of the ABA methodology and functional behavioral assessments. Section 5 presents the proposed ABA ICT service platform, providing a detailed description of the main identified ICT technological components needed for the development of an advanced system for supporting the ABA model’s application. In Section 6, a reference use case, which can be considered for the platform’s validation, is defined, highlighting how the proposed solution is beneficial with respect to some of the already existing and adopted technological tools. Challenges for enhancing the ABA model’s application by means of ICT capabilities are considered in Section 7. Finally, conclusions are drawn in Section 8.

## 2. Neurodevelopmental Disorders and Social Inclusion

In order to investigate how ICT may support the social inclusion of children with disabilities, it is important to characterize the application’s context by highlighting both what neurodevelopmental disorders mean according to recent studies and what are the main features and purposes of social inclusion.

### 2.1. Neurodevelopmental Disorders

The publication of the *Diagnostic and Statistical Manual of Mental Disorders, Fifth Edition* (DSM-5) [2] represents a particularly significant event from which some issues deserve to be explored. DSM-5 replaces the “Disorders Usually First Diagnosed in Infancy, Childhood, or Adolescence” section of DSM-4 [3] with a new chapter titled “Neurodevelopmental Disorders”, which includes intellectual disability (intellectual developmental disorder), communication disorders, autism spectrum disorder (ASD), attention deficit hyperactivity disorder, specific learning disorder, and motor disorders. This reflects a general approach in DSM-5 that is be representative over a lifespan [4].

The “Neurodevelopmental Disorders” chapter replaces the outmoded term *mental retardation* with *intellectual disability* (intellectual developmental disorder), a term that has come into common use over the past two decades among educational and other professionals and advocacy groups.

DSM-5 defines the levels of severity based on adaptive functioning rather than IQ (intelligence quotient) scores, because it is adaptive functioning that determines the levels of needs and necessary supports. Moreover, the term *attention deficit hyperactivity disorder* was included in the new category of neurodevelopmental disorders, and some major changes were made regarding autism diagnoses [5].

In order to arrive at a diagnosis of intellectual disability, three criteria have to be met. These are as follows:*Deficit in intellectual functions*: DSM-5 particularly underlines deficits in the following:
-Reasoning;-Problem solving;-Planning;-Abstract thinking;-The ability to judge;-School learning;-Learning from experience.These deficits are to be confirmed both by a clinical evaluation and individual and standardized intelligence tests.*Deficit in adaptive functioning*: This results in the failure to achieve developmental and socio-cultural standards relating to both autonomy and responsibility. These are adaptive deficits that, in the absence of constant support, limit functioning in one or more activities of daily life, such as communication, social participation, and autonomous life in various living environments of an individual (home, school, work environment, and community).*Outset of the first two points during the development period*.

As other neurodevelopmental disorders, intellectual disabilities are also characterized by frequent co-morbidity (e.g., ASD). Moreover, DSM-5 suggests specifying the severity of intellectual disability, abandoning the reference to the scores of intelligence tests. In the framework of severity levels, the term “Extreme” takes the place of “Very severe”. Therefore, the defined severity levels are now indicated as mild, moderate, severe, and extreme [6]. Another important change to underline within [2] concerns the introduction of the definition of ASD. This definition includes the following: the DSM-IV autistic disorder, Asperger’s disorder, childhood disintegrative disorder, pervasive developmental disorder, and not otherwise specified. People who had previously been diagnosed with one of these diagnoses can now meet the criteria for ASD, while people with deficits in social communication whose symptoms do not meet criteria for ASD should be evaluated for social communication disorders [2]. However, this change, motivated by evidence-based research, has raised international concerns. Changing the criteria for diagnoses may have some negative effects. For example, it may affect the number of people involved in the future, with the risk of penalizing those who, failing to meet the new diagnostic criteria, could lose the right to health, school, and social support. ASD is a generic term used to describe a set of neurological development conditions that are characterized by difficulties in verbal and non-verbal communication, imagination, and social interaction; and by repetitive or limited interests and behaviors that are evident during early childhood [2]. Already from the first year of life, in fact, there are some signs relating to inconsistent behaviors, as well as absent behaviors. Examples include a lack of eye contact; the orientation to name; stereotypes and self-stimulations; difficulty in managing and understanding emotions; resistance to changes; and imitation [7,8]. The diagnostic criteria recognize that for some children, the difficulties may not emerge until a later age when social needs will go beyond their skills [2]. The characteristics of ASD vary from person to person, both in terms of the severity of behaviors and the combination of symptoms. No two children behave the same way. As a spectrum, ASD has numerous and different characteristics, and the level of impairment of skills can range from a mild (highly functional autism) to severe degree. As soon as it is diagnosed, the child with ASD together with his parents can refer to some effective treatments (for example, in the Conclusions, we will refer to the Early Start Denver Model) to minimize the impact of the impairment of skills with the aim of not only improving the quality of life, first of all, for the person with ASD but also for their family members [2,9,10].

In March 2022, the DSM-5-TR version was released [11]. Thanks to the study of four groups of reviewers that were engaged in transversal issues (culture, sex and gender, suicide, and forensic medicine) and a working group committed to the use of fair and non-stigmatizing language, numerous changes have been made to the previous DSM-5 version. In particular, the new diagnosis of Prolonged Grief Disorder (F 43.8) was introduced, and significant changes were made with respect to the diagnostic’s conceptualization and the diagnostic criteria of over 70 disorders. For the purposes of this contribution, we must highlight that in DSM-5-TR’s criterion A, which relates to the diagnosis of autism spectrum disorder, has been revised and made clearer [12,13].

We also consider that it is important to introduce the construct of quality of life (QoL) within this paper. This construct represents a significant aspect on the lives of people with and without disabilities. Among the several models focusing on the multidimensionality of the quality of life, the eight-domain model by Schalock and Verdugo [14] is the one most commonly cited and used in intellectual disability research. The eight main domains of this model comprise material wellbeing, physical wellbeing, emotional wellbeing, social inclusion, personal relationships, self-determination, personal development, and rights.

As many authors have affirmed [15,16] in the last thirty years, the construct of QoL, in addition to becoming increasingly important in the planning of individualized interventions and programs and in organizing services, has contributed to the change in the representation of people with disabilities, affirming how the quality of life should become the main pursuit. We additionally need to add that the domains and the corresponding indicators of the QoL construct are culturally sensitive, connected to current and future political questions, and are under the control of people with disabilities, together with their personal and organizational networks [17].

### 2.2. Social Inclusion

As proposed in [18] social inclusion is a broad term that includes social interaction and community participation. With social interaction, the authors refer to relatives, colleagues, friends, acquaintances, and intimate partners (with or without disability). With community participation, they refer to leisure activities (hobbies, art, and sport), productive activities (employment or education, consumption or access to goods and services, etc.). Social interactions and participation in the community tend to be influenced by a person’s characteristics, age, gender, communication, and language skills; economic and time resources; and ability to move and adapt to potential changes. Social inclusion, by placing the person at the center of reflection—together with one’s expectations, needs, strengths, and weaknesses—is a process that strongly involves aspects of community life, institutions, and policies. Each person has their own vision, and a more or less complex idea of this process is as follows: the vision is dynamic, unstable, and in continuous construction, where being included is not tied to a prescriptive role, a norm, or a constraint but implies a continuous structuring and destructuring of organizations, institutional, and social contexts, and attention gives a voice to those who experience and live within it [19].

The analysis of social inclusion for children with neurodevelopmental disorders cannot be considered a specific topic that is related to how someone can be integrated in a specific place (community, family, school, etc.); it should indicate a perspective toward modifying cultures, organizational forms of contexts, and relational modalities so as to be able to respond to requests for learning and participation. Promoting inclusive paths and processes for children with disabilities therefore becomes a systemic and across-the-board action, and a school should be a privileged location for comparison, research, and study.

The UNESCO views inclusion as “a dynamic approach of responding positively to pupil diversity and of seeing individual differences not as problems, but as opportunities for enriching learning” [20]. This involves a willingness to accept and promote diversity and to take an active role in the lives of students both in and out of school. The optimal learning environment for inclusion depends largely upon the relationship among teachers, parents, other students, and society. Ideally, effective inclusion involves implementation both in school and in society at large. Social inclusion therefore means respecting and enhancing differences, promoting meaningful and functional relationships within school and extra-curricular activities.

According to [21], inclusion involves change. It is an unending process of increasing learning and participation for all students. It is an ideal to which schools can aspire toward but that which is never fully reached. However, inclusion occurs as soon as the process of increasing participation is started. An inclusive school is one that is on the move. Taking part does not only mean learning together with others or collaborating with them in shared learning experiences. Taking part means having a voice on how education is proposed, implemented, and experienced, and even more deeply, taking part means being recognized, accepted, and appreciated for who we are, for what we think, and how we communicate.

In schools, as in society, activities and actions that tend to welcome, listen, know, and enhance a person with respect to their characteristics and differences must be placed in relation with didactic, normative, organizational, relational, and spiritual aspects. Training becomes an aspect and, at the same time, an objective that should be constantly promoted and stimulated across-the-board and in an interdisciplinary manner.

Being familiar with and continuing research on how the mind of children with autism works, for example, help promote the previously introduced quality-of-life concept. As highlighted in [22], there are two aspects that make the life of a person with autism difficult. The first is made up of biological constraints that make an individual’s brain less suitable for social learning. The second comprises ignorance, which still remains with respect to autism. Knowing how the child and their mind works, as well as applying the appropriate treatment, can make a difference in their life. It is true that the child’s genes contain substantial information, but with respect to whether the child will be happy, whether he will be able to develop his full potential, and whether he will participate in a game with understandable rules, these aspects are not written in the genes.

The theme of social inclusion represents an area of interest and reflection that is particularly present within both the United Nation’s Convention on the Rights of Persons with Disabilities (CRPD) [23] and the International Classification of Functioning, Disability, and Health (ICF) [24]. These international documents, in addition to having already triggered profound changes on the quality of life of millions of people with disabilities in the world, will continue to promote cultural, political, and social transformations, building new approaches and knowledge, transforming all sectors of society, and fueling the systemic action called social inclusion. The same documents influenced the contents of the 2030 Agenda [25] with its 17 Goals for sustainable development. Within the Agenda, issues relating to people with disabilities have found a new central role in international development strategies. In many Agenda Goals, direct references are made to people with disabilities. Within Goal 4, for example, titled “Ensuring inclusive and fair quality education and promoting lifelong learning as an opportunity for all”, in addition to the commitment to eliminate gender disparities in education by 2030, we can find a commitment to ensuring equal access to all levels of education and vocational training for all vulnerable groups, including people with disabilities. Furthermore, the Agenda refers to building and updating educational support measures that are sensitive to minors, disabilities, and gender and also refers to offering a safe, inclusive, and effective non-violent environment for all. It is also important to highlight that within the Agenda, there is an interdependence of the objectives and the related targets, with particular reference to education, health, violence, especially gender, emergency, accessibility, and training.

In accordance with the “Universal Design” strategy in article 2 of the CRPD [23], the Agenda ultimately invites the following:-Design products, structures, programs, and services that are accessible and usable by all people to the greatest extent possible and without any adaptation or specialized designs (approach that does not exclude support devices for particular groups of people with disabilities where necessary);-Use appropriate national and international standards regarding the accessibility and usability of new technologies (accessibility of websites, workstations, books and electronic documents, etc.), taking into account the economic and technological differences existing in developing countries [26].

## 3. The Model of the Applied Behavior Analysis

The theoretical foundations of the ABA approach present clear connections with Skinner’s theory of operative conditioning [27]. In a nutshell, we can encourage a specific behavior by rewarding it systematically and at the same time discourage another behavior by punishing it systematically [22]. In accordance with [28], ABA is a research program aimed at identifying the factors that motivate behavior, such as adaptive or maladaptive and adequate or inadequate behavior, in a given context (e.g., at home, at school, etc.). ABA focuses primarily on socially significant behaviors that can be measured and quantified, allowing for the constant monitoring of a child’s behavior. By carrying this out, in addition to assessing whether there are improvements, one has the possibility to estimate whether the adopted procedures are appropriate for that particular child in that specific context [29]. ABA is therefore a set of methods, procedures, and techniques that, in finding cognitive and experimental references in behavioral sciences, is based on evidence, constantly resorting to the methodology indicated by scientific research and placing the empirical validity of its own results in focus. This aspect represents one of the strengths of ABA, and it is worthwhile to investigate this aspect further. Via the constant collection of data within structured learning sessions, an ABA therapist (or ABA tutor) under the guidance of an ABA supervisor traces a baseline relating to a specific behavior. Thereafter, by implementing the procedures and checking whether the behavior has changed or not, compared to the previously traced baseline, the therapist and supervisor contribute to an objective monitoring of the ABA procedure as a whole. All this allows a guarantee that the ethical terms from research studies are based on scientific evidence. In operative conditioning, a dominant theme in the ABA approach is reinforcement: In particular, attention is focused on identifying the most effective reinforcements for a child and for a person. Reinforcements that are more effective are those linked to the child’s (person’s) interests and special educational needs and those that increasingly tend to promote learning as more than a punitive component. In this regard, by trying to describe a typical ABA session, within a structured context, we can imagine a child that is physically guided by an adult in performing a certain desired action. The child will be rewarded when he completes the action, and in subsequent sessions, help from the adult will gradually decrease until the child has mastered that specific skill [22]. Reinforcements—defined by Skinner [27] as events that, following the emission of a certain behavior, make the behavior more likely to appear in the future—belong to different dimensions (material, social, sensory, symbolic, etc.) and can respond to different strategies (continuous reinforcement, and intermittent reinforcement) and different methodologies (reinforcing immediately after the emission of a certain behavior, progressively replacing material reinforcements with more natural reinforcements, etc.). Within this study, in which we propose some strategies on how ICT can improve the application of the ABA model in the life contexts of children with neurodevelopment disorders, it is useful to refer to the connection between the ABA approach and natural contexts. As highlighted in [30], an ABA intervention pursues the best results when it is carried out in environments that are significant for the person involved. The school context and home environment represent privileged locations that can trigger learning for a child. At the same time, school and home are also potential contexts where inappropriate behaviors can occur with important frequency and intensity. In connection to this consideration, we must highlight how programs based on parent training and teacher training are provided within the ABA approach. In parent training, family members become a real training object, and the aims include the following: defining a problem by studying the problem with professionals, methods for collecting useful information, and which alternative solutions to experiment with and what methods to use in verifying them. In this sense, the ABA approach welcomes and gives meaning to the concepts of co-participation, the co-construction of the life project, psychoeducational alliance, and compliance [31]. Furthermore, teaching methods, procedures, and intervention techniques and educating family members mean assigning them an active role, which, as suggested by research, does not create stress for the family but tends to renew the enthusiasm toward the child. On the contrary, stress in the family increases when parents are not provided with tools, indications, and concrete information on what actions to take [32]. Regarding teacher training, some general considerations are necessary. ABA-based treatments have shown clinically significant improvements inherent in intellectual, social, emotional, and adaptive functioning in the presence of neurodevelopmental disorders. In particular, its systematic application with people with ASD and people with intellectual disabilities has highlighted improvements in socially useful behaviors, communication, and language [33,34,35,36,37,38]. The importance of early and intensive intervention was highlighted already in the last century in numerous further studies [39,40,41]. As pointed out in [42], ABA also has several key features:-An emphasis on positive reinforcement procedures to build behavioral repertoires;-Functional evaluation of individual behavior;-The use of scientific methods to evaluate the effects of interventions;-The identification of teaching objectives and procedures;-The progression from simple to more complex skills;-The transfer of instructions from structured contexts to natural contexts;-The training of family members and other people involved to implement interventions in different contexts.

Finally, the use of ABA is the only educational method that has resulted in significant improvements in the main deficits of ASD [34,36,43,44], and it has social validation provided by family members who use ABA at home and who appreciate its positive impact on the life of the entire family.

Despite the demonstrated effectiveness, the ABA approach faces difficulties in terms of its development in school settings. The reasons for this are different. For example, it is important to reflect on the nature (public or private) and on the type (mixed or special) of educational institution. Similarly, it is important to consider what the political, educational, and health choices of the institution are and whether the ABA approach in that institution has obtained regulatory recognition. In addition, the difficulties of development in school environments may also depend on the personal characteristics of teachers in terms of whether they are available to train themselves and on the availability of qualified professionals in that particular context. A transversal element across these different situations is the cost of the ABA approach due to its intensity (number of hours per week) and its organizational characteristics (from two to three ABA therapists; tutors, supervisor, and technical assistance). Teacher training can be considered expensive, but dependence on external consultants who intervene at a school and provide comments, assessments, etc., is also expensive. As highlighted by [45], teachers generally report a dissatisfaction with this external consultancy model, and they invite consultants to spend more time in the classroom in order to provide practical strategies. Linked to this aspect, one weakness of the ABA approach is represented by the technicality of the language that distinguishes it. Technicalism that becomes an obstacle in networking because the language used is not simple or intuitive.

With regard to the intervention techniques in the ABA approach, the most relevant ones can make references to the following strategies [46].

*Aid techniques and aid reduction*: Gestural, physical, and verbal suggestions (prompting) are provided to guide the child in learning adaptive behaviors. The progressive attenuation of the aid (fading) is achieved by reducing both the intensity and frequency of suggestions and is oriented toward maintaining the achieved adaptive behaviors.

*Imitative learning techniques*: These techniques are based on imitative learning (modeling) and demonstrate a significant presence in ABA interventions. They are models of behavior that the child is invited to reproduce. Correct imitation corresponds to verbal or gestural reinforcements, as well as the possibility of accessing preferred objects or activities. Reinforcements are used to encourage the maintenance and generalization of learned behavior.

*Modeling techniques*: Modeling techniques (shaping) allow the development of new skills, starting from those already acquired. Shaping is the process of differentially reinforcing successive approximations toward a desired response [47]. These techniques represent central elements of both the ABA approach and the models that are derived from it, such as the discrete trial training model, pivotal response training model, and Denver model [48,49,50].

*Linkage techniques*: Chaining is a strategy used for teaching complex skills consisting of sequences of well-defined behaviors (for example, the individual parts of a complex ability, such as dressing). To carry out chaining, it is necessary to identify a procedure that begins with the division of the skill into individual components (task analysis), continues with the construction of the behavioral chain, and ends with the structuring of a program by chaining the components via step-by-step reinforcement [51]. Within the techniques described and more generally during the ABA sessions, numerous data are acquired and recorded. This information must be shared with all those involved in the team (from two to three ABA therapists; tutors, supervisor, sometimes a senior tutor, and teachers) and with the child’s family members. This expanded sharing is fundamental to obtaining the best results since the program must be constantly followed both at home and at school. However, the information is collected on paper, which, in addition to requiring organizational efforts, represents a potential source of errors [29].

## 4. ICT as a Support for the ABA Model

The adoption of ICT is widely recognized to have a high potential to provide opportunities for social inclusion [52]. Specifically considering children with disabilities, technological tools can be used for different purposes, and they range from education (school e-inclusion) [53] and rehabilitation [54] to playful and entertaining activities [55]. Motor, cognitive, sensory, communicative, learning, social, and leisure skills are some of the main skills that can be improved via the exploitation of technology in different activities, including playing and entertainment.

Focusing on social inclusion for children with neurodevelopmental disorders, ICT can first contribute to improving children’s relationships within proximate contexts (family, schoolmates, and group of friends) and facilitate those in distal contexts (public parks, entertainment venues, and community initiatives).

In detail, aiming at analyzing the child’s behavior in order to improve targeted skills via the adoption of ABA therapy, the technological support provided by ICT appears to be straightforward. In fact, the ABA model strongly depends on evidence-based strategies: the behavior has to be observable and measurable, and the decisions are always carried out by relying on experimental data.

The objective of this section is to provide a high level description of how the ICT can be exploited to enhance the application of the ABA model by highlighting their impact on the methodological process and, in particular, on functional behavioral assessments. Moreover, in order to better introduce the proposed ABA ICT service platform, the existing link between ABA and ICT is described together with the capabilities and potentialities of the new technological tools that need to be considered for the definition of innovative and advanced solutions for ABA.

### 4.1. ICT for ABA Methodology and Functional Behavioral Assessments

#### 4.1.1. ICT for ABA Methodology

The key element that makes ICT solutions suitable for the application of the ABA model is that ABA is an evidence-based approach that relies on data analysis via the individualized investigation of the “how” and the “why” of a particular behavior. From a technological point of view, the ABA is a conceptual and systematic approach that analytically evaluates the observable and measurable data associated with a behavior. The assessment of measured data should lead to a decision process with the objective of modeling a behavior to achieve a targeted skill.

In order to identify the main potentialities of ICT, it is worth giving an overview of the main ABA methodological components:Task analysis: This comprises breaking down a skill into smaller and manageable tasks by identifying the targeted skill, its prerequisites, and its task components. After testing and teaching the skill, an assessment is carried out in different situations.Shaping: This represents the act of gradually modifying the child’s behavior to the desired one.Generalization: This consists of the application in real-life contexts of what the child learned in class.Prompting: This aims to redirect the child’s attention and behavior by using gestural, verbal, positional, visual, and physical inputs.Fading: This comprises decreasing the amount of assistance required by a child for completing an assigned task.Extinction: This is based on ignoring the reinforcement of problematic behaviors.

All the aforementioned components can take advantage of the primary features provided by ICT services and systems: customization, automation, monitoring, communication, real-time assessment, and validation.

Each component of the ABA methodology can be adapted to the child via a personalized intervention in order to speed up the behavioral modeling process, and each component used is more effective in achieving the target skill.

An automatic process can guide the basic teaching strategies along with the ABA methodological components in order to define a targeted and optimized intervention. A continuous monitoring system can be used for a remote and real-time assessment of the behavior in different time frames: before, during, and after a selected teaching strategy. The data evaluation can activate a decision-making process to modulate the teaching strategy, which will be tested and validated in daily-life environments.

According to these considerations, a list of actions that highlight how ICT can positively impact ABA methodology is reported in the following:-Provides highly personalized interventions based on the child’s profile (e.g., age, type and degree of disability, behavior needs, and family requests);-Provides computer-based tools to support and automate teaching strategies along the methodological components of the ABA model;-Monitors the child’s behavior using advanced sensor systems for the automatic and real-time gathering of data related to emotions, physical activity, and reactions to specific events;-Provides a real-time assessment of the gathered data to accelerate the intervention’s decision making according to the analytic evaluation of the collected data;-Validates the teaching strategy for optimizing the intervention either in class or daily-life environments.

#### 4.1.2. ICT for Functional Behavioral Assessments

A fundamental component at the basis of the ABA is functional behavioral assessments, and their primary objective is to answer four important questions related to a behavior: *why, how, where, and when does that particular behavior occur?* The ability to properly answer to those questions will influence the effectiveness of ABA therapy to produce positive changes in behavior.

A pillar of the ABA is the antecedent–behavior–consequence assessment. This involves the description of the occurrences of the target behavior related with the events occurring immediately before and after the behavior itself.

The why of a behavior can be related to a combination of antecedents that cause the behavior itself. However, the identification of the antecedents is commonly difficult to perform, particularly when dealing with children with neurodevelopmental disorders. Let us try to imagine the complexity in recognizing an antecedent, which in this case refers to an autistic child whose problematic behavior is generated by physical malaise.

In order to fully define the process of functional behavioral assessments, the reason for the occurrence of a behavior has to be associated with the way it looks and to the environmental context where it takes place and when it occurs.

There are some cases in which the functional behavioral assessment cannot be simply defined due to the lack of antecedent recognition and the impossibility of properly monitoring the behavior. Some antecedents that are the cause of stress cannot be recognized because they commonly do not produce any visible consequences during the observational study performed by the ABA therapist. Moreover, a proper assessment of the frequency, intensity, and form of a problematic behavior requires accurate monitoring for a long period of time. This is not always achievable in noncontinuous observational studies.

In this context, ICT can be exploited to support the functional behavioral assessment of the ABA model using remote and real-time context-aware behavioral analyses.

### 4.2. Background: ABA and the ICT Existing Link

One of the first evidence of a link between ICT and ABA dates back to the 1960s. Researchers Ferster and DeMyer [56] created an experimental room that contained a large number of devices that, when operated by either a coin or direct key, provided some rewarding consequence for the child. The room contained a one-way vision screen on the wall facing the experimental devices. The study involved four children: an 8-year-old boy and a 9.5-year-old girl with autism and two matched regularly developing controls. The study used reinforcements such as food, including candy, trinkets, and coins. Ferster and DeMeyer noted that the skills once taught in a controlled laboratory setting could then be used to “investigate many aspects of the autistic repertoire which have heretofore been inaccessible”. The study proved to be a success. Ferster and DeMeyer’s study was soon followed by the rapid establishment of ABA as a credible treatment and assessment technology.

Starting from the late 1970s, the advances in information technology allowed developing increasingly functional software for improving the learning opportunities and conditions of people with neurodevelopmental disorders [57,58,59,60]. Within this context, one research area focused on developing software for simplifying the learning of new topics via the use of a computer [61]. Another research area focused on mechanisms that improve the interactions between people and their environment. This period was also characterized by the emergence of studies connected to augmented communication [62] and to the use of robotics [63]. In this regard, Alcade et al. [64] observed that computer-assisted learning can be an efficient learning–teaching procedure for children with neurodevelopmental disorders. A specific software called “Let’s Play With…” was designed to teach the concepts of colors, shapes, and body positions.

From 2000 onwards, studies expressing the importance of the connection between ICT and ABA became increasingly commonplace. The following devices and systems are some examples: computer-aided system [65,66,67,68], video modeling/video prompting/video conferencing [69,70,71,72,73], optic micro-switch [74], picture exchange communication system (PECS), and voice output communication aids (VOCAs) [75,76,77].

In the last decade, ABA experiments have increasingly involved persons with severe/profound multiple disabilities, and among the most commonly adopted ICTs, it is worth mentioning the following: microswitches, VOCAs, and microswitch clusters [78,79,80,81,82,83].

We can witness the continuous development of research and studies with respect to the use of computer-assisted instruction and computer-based interventions [84,85], the use of speech-generating devices (SGD) in communication interventions [85,86,87], and the strong increase in the use of software applications via mobile devices (i.e., smartphones or tablets) [88,89,90,91,92]. In addition to supporting ABA programs in the dimensions of learning and relationships, ICTs demonstrate an increasing presence in health-related studies [93,94,95].

### 4.3. New ICT Potentialities for ABA

In recent years, a remarkable technology that has introduced great impacts on ICT, which can be also a promising support for ABA with respect to neurodevelopmental disorders, is artificial intelligence (AI) [96]. This technology aims to emulate classical human intelligence processes, including learning, reasoning, and making decisions based on the experience. AI is applied in different application contexts, including business [97], healthcare [98], education [99], autonomous vehicle [100], and robotics [101]; it is often exploited to enhance other technologies, such as speech recognition, natural language generation, robotic process automation, and text analytics. A practical use case of AI involving behavioral analyses can be observed in the commerce sector, and it is termed customer behavior analysis [97]. In this case, customer behavior analysis aims to learn individual behaviors in order to personalize products and improve selling strategies. In the context of the social inclusion of children with neurodevelopmental disorders, the personalization feature is crucial in identifying the proper teaching strategy that can enable a child with special needs to reach a target skill. Several studies have shown how AI has been used to analyze and shorten the behavioral diagnosis of neurodevelopmental disorders, such as autism [102,103]. Many mobile software applications for smartphones and tablets have also been developed for aiding ABA therapy. Most of them refer to providing support within a particular ABA method, such as PECS, using prompting procedures or step-by-step functional analyses with respect to the antecedent–behavior–consequence model. However, the present use of AI for ABA in neurodevelopmental disorders remains restricted to diagnoses or to tracking the child’s behavioral progress during ABA therapy. Moreover, AI algorithms process the data that are manually collected by therapists during therapy sessions. Although they can be supported by specific apps for behavioral tracking, the management of this operation is not always easy. An advanced ICT architecture—which exploits both the automatic collection of some behavior-related data performed by the adoption of wearable and non-wearable sensors and the AI algorithms that process them in real time during the different steps of ABA therapy—is missing. A type of ICT architecture that allows the feasible continuous monitoring of a child’s behavior not only during the training session but also during the child’s daily activities is needed, as it can provide therapists with precious information for understanding the child’s progress across different ABA methodological components.

## 5. The ICT Service Platform for ABA

The adoption of ICT can be a promising method for gathering different data related to the child’s behavior, including data that cannot be obtained using classical observational studies. The possibility of continuously collecting behavioral data in real time can be essential for functional behavior assessments: It would allow the monitoring of a child for an entire day in different contexts, such as the domestic context. Behavioral analysis usually provides training for family and caregivers since the child spends most their time with them. Providing a system that is able to transparently and automatically gather data to be postprocessed by behavioral experts may help them to better make decision with respect to ABA therapy applications. Moreover, carrying out data processing for automatic and customized behavior analyses based on the child’s profile may support therapists during the different phases of ABA therapy interventions. This can be achieved via the adoption of AI algorithms that are opportunely trained using the collected data.

Figure 2 depicts the high-level design of the reference ICT service platform proposed for supporting the application of ABA methodology. It aims to provide an overview of the identified solution by highlighting the main components and interactions among the entities involved in the system.

To better describe the overall functioning of service, which aims to improve the efficiency and effectiveness of the application of the ABA model and to introduce the technical details provided in the following subsections, a brief high-level description of the main components required for the implementation of such a service is reported. In particular, the three main identified technological components are as follows:-*Data acquisition component*: The data acquisition component is represented by a monitoring system comprising both environmental and wearable sensors. They allow collecting multiple data sources (e.g., bio-metric indicators, physical activity parameters, sounds, and videos) for processing in order to study and evaluate the child’s behavior before, during, and after ABA therapy treatment. The sensors enable the gathering of a large amount of information needed for the child’s behavioral analyses and allow the creation of datasets that can be used for the automation of the ABA model’s application.-*Data processing component*: The data processing component is represented by the AI engine, which is the core of the proposed system and is part of the ABA ICT service center’s platform. It provides real-time highly personalized guidelines of interventions based on the child’s profile (short-term analysis), and it helps therapists when defining the most suitable ABA program by considering background experiences (long-term analysis). The AI algorithms are opportunely trained with adequate datasets coming from both the monitoring sensor system and from the already available experience-based data collected by the therapists during their experience in the application of the ABA model.-*Data communication component*: This component is represented by the telecommunication network’s infrastructure, which supports the data exchange among the network nodes involved in the system’s architecture.

The overall functioning of the proposed system can be summarized as follows. The child’s emotion-related indicators and environmental surrounding data are transparently and continuously collected by an advanced sensor monitoring system. These data are transmitted to the ABA ICT service center platform for permanent storage and processing based on their source and the child’s profile. The AI algorithms are responsible for the processing of the data and for providing the key elements as outputs in order to ease and improve the efficiency of the application of the ABA model (as better detailed in the following subsections). ABA teaching strategies (including personalized programs), short- and long-term behavioral assessment, and functional behavioral analysis (ABC) are some of the output of the application of the AI algorithms on the collected data. ABA therapists can access this information in real time and are supported in their activities, and the therapists also represent a source of information for the AI engine. In fact, they can provide the system with additional data, which are then considered using processing algorithms for the evaluation of specific situations. The guidelines provided by the proposed system can be classified based on the different contexts that the child has experienced (learning, family, and community). Moreover, some advice directly coming from the system can be forwarded to the child’s family members in order to help them in the management of difficult daily situations. Figure 2 highlights the main components and interactions among the entities involved in the system, providing an overview of the referenced ABA ICT service platform, which aims to improve the efficiency and effectiveness of the application of ABA methodology.

The main aforementioned identified technological components needed for the development of the proposed ICT service platform are described in detail in the following subsections.

### 5.1. Data Acquisition Component: Sensors for ABA Measurement

The ABA model can take advantage of data collected using both wearable and environmental wireless sensors, as illustrated in Figure 3.

Wearable sensors generally refer to body-worn technological devices that are often available for everyday worn items, such as belt, watch, glasses, necklace, and shoes. These sensors can provide information about sensory perception, biomedical parameters, and motion activity. This information can be helpful in evaluating the child’s feeling in a particular context and to assess the resulting behavior.

*Perception sensors* [104,105] can catch the visual and auditory perceptions of the child, providing information about what the child is watching or listening to, e.g., recording verbal/vocal interactions and images using sound and camera sensors, respectively.

*Biomedical sensors* [106,107] provide parameters such as skin sweat, heart rate, body temperature, blood pressure, and body temperature, which can be used to generally evaluate the child’s health or, in particular, to detect malaise circumstances, which could be the cause of a problematic behavior that is not visible by only carrying out observational studies.

*Motion and location sensors* [108,109,110] refer to inertial sensors that are placed on a specific body position, and they are capable of detecting body motion. These data can be processed in order to recognize particular gestures or, more generally, other body poses or motion activities, such as seating down, lying down, standing, walking, running, etc. Location and positioning sensors can be used to determine the child’s location both indoor and outdoor.

The information coming from visual and auditory sensors can be exploited to investigate the prospective causes of a problematic behavior or, more generally, to relate input information received by a child that resulted in certain behaviors. On the other hand, the data provided by biomedical and motion activity sensors can be combined to assess the resulting behavior.

*Environmental sensors* [111,112] are external devices that provide contextual information, and they can be located in either indoor or outdoor areas. The environment can strongly interact with the child’s feeling, and behavioral analysis cannot disregard a conscious consideration of the environmental context. These sensors can be used to determine physical ambient parameters, such as temperature, light brightness, and humidity, or to detect acoustic noise and perform video analyses.

Some of the main monitored parameters that point to problematic behaviors are as follows: frequency, rate, and duration. Therapists use different methods to collect data; however, in order to properly assess a behavior (analyzing the frequency, intensity, and the form of a behavior), the continuous real-time monitoring of the child is required. Observational studies only partially meet these requirements: they are not continuous and do not always last for a long period of time due to several logistical reasons.

The data provided by wearable and environmental sensors may need to be pre-processed and/or aggregated to improve the quality of data and to limit the quantity of data eventually transmitted to an ICT service platform, supporting the real-time context-aware ABA system.

### 5.2. Data Processing Component: AI for ABA

Artificial intelligence (AI) [96] may play a fundamental role in the implementation of behavioral analyses. AI consists of hardware and software components that are able to learn information based on training data, similarly to how human beings learn by imitation. Actually, AI systems do not only imitate, but they aim to generalize what they learned. This kind of generalization requires substantial data in order to properly carry out the learning process. In the era of the Internet of Things, almost every commercial device is equipped with sensors that are able to provide different types of data. This is one of the primary reasons why AI has currently become very popular [113]; the possibility of obtaining a large amount of data enables AI algorithms to increase their performance in terms of their ability to learn.

For the ABA, AI could be exploited to recognize human emotions [114] and gestures [115,116] or to learn appropriate methods for breaking down a target skill that refers to a particular intellectual disability. According to the target skill and the child’s profile, AI algorithms can be trained using the data provided by monitoring systems in order to automatically evaluate behaviors and suggest how methods for modifying the teaching strategy. AI can also be used to define the suitable contexts for shaping and generalizing behavior or to determine prompting actions, including prompting a decrease in steps based on the child’s progress.

The effectiveness of AI algorithms relies on the availability of real datasets. Data should be collected anytime and anywhere from controlled environments, such as classes and daily life environments (e.g., family context). To this aim, advanced wearable and environmental sensor systems for real-time data collection and monitoring are crucial for feeding data to AI algorithms.

Once AI algorithms have been trained using historical data and the AI system receives the target skill, the child’s profile, and the real-time data collected using wearable and environmental sensors as inputs, as depicted in Figure 4 and Figure 5, the AI system should autonomously provide a strategy that defines the most suitable approach to ABA methodology:Task decomposition into smaller tasks;The differential reinforcements for shaping the desired behavior;The prompting types among gestural, physical, verbal, visual, and positional prompts;The fading schedules for the identified reinforcements;The list of reinforcements for problematic behavior in order to achieve the problematic behavior’s extinction.

The candidate AI algorithms for behavioral analysis are based on supervised [117] and unsupervised [118] machine learning techniques. Supervised learning, which relies on example data in order to classify behaviors, includes a number of classification methods: deep learning using neural networks [119], decision tree, support vector machine, and k-nearest neighbor [116], each of which presents different characteristics in terms of memory usage, prediction speed, and predictive accuracy. On the other hand, unsupervised learning can be exploited either to group behaviors based on particular features or to detect anomalies in behavior.

### 5.3. Data Communication Component: Tele-Communication Network Infrastructure

The functioning of the overall service platform relies on secure and reliable telecommunication network infrastructure, which enables data exchange among different network nodes. It consists of the following:-Short communication links: They allow communication between environmental and wearable sensors and the data collector and gateway. These may include near-field communication, Bluetooth Low Energy, and WiFi links.-Long communication links: They enable communication between the data collector and the gateway and the ABA ICT service center platform. Moreover, they allow end-users to access the service.

Figure 6 depicts the ICT network architecture that allows supporting the proposed ABA service. In particular, the main network nodes responsible for data acquisition, processing, and transmission are highlighted with the communication links among them. It is worth observing the role of the data collector and gateway, which are responsible for the following: (i) the gathering of data coming from wearable and environmental sensors via different communication network protocols (WiFi, BLE, and NFC); (ii) the pre-processing of these data for reducing the amount of data to be transmitted over long-range communication links; and (iii) the transmission of pre-processed data to the ABA ICT service center platform using 4G/5G and/or wired network access technologies.

In order to highlight how end-users access the platform, Figure 6 also shows remote monitoring devices with user interfaces. Therapists and the child’s family members can interact with the platform using these devices and receive suggestions (output of the AI engine) or provide additional information to enhance the AI output based on the child’s specific needs and profile.

## 6. Application Context: Use Case

This section aims to highlight how the proposed ICT platform could support the application of ABA therapy. A detailed definition of a reference use case describing the main activities carried out by a therapist during an ABA training session is provided. This use case is considered for the platform’s validation once the system has been developed. Moreover, the main benefits and innovative features of the proposed solution are specifically analyzed by referring to the identified use case and by carrying out comparisons with respect to some already existing and adopted technological tools.

### 6.1. Definition of a Reference Use Case

The initial goal of an ABA treatment is to create a trusting, fun, and motivating relationship between the child and the therapist (pairing). In the first phase, the therapist does not devote himself to working on specific therapeutic objectives, but by playing with the child, he tries to know the child’s interests and favorite stimuli. Once that phase is over, the real work begins. One should always try to personalize this work because each child has specific characteristics with respect to functioning. The ABA session can be presented using two different approaches. On the one hand, there are activities that take place in a highly structured context (for example, a table), are aimed at a specific task (matching images, indicating objects, learning to make sounds, etc.), and involve a closed interaction between the child and the therapist. This method is called discrete trial training (DTT). On the other hand, there are activities that are carried out in a more natural context (for example, a playground), activities that start from the analysis of the child’s favorite stimuli, and are aimed at achieving useful everyday life skills (playing with peers, socializing with unknown people, having a role within a specific phase of a game, etc.). This method is called natural environment training (NET). Next, we attempt to find out what happens during DTT. The session begins with a discriminative stimulus (SD) or rather an instruction that is provided by the therapist to the child. SD can be verbal but also non-verbal (use of images, toys, objects, etc.). To be effective, this instruction must be clear and coherent, free of unnecessary information, and supported by a slightly higher tone of voice (by the therapist) compared to normal use. Following the instruction, the therapist has two possibilities: (i) to wait for the child’s response; and (ii) alternatively, to provide help via a suggestion (prompt) if necessary, which comprises a form of assistance that facilitates the correct response from the child. This suggestion will be gradually removed by the therapist following the achievement of adequate skills by the child. Finally, following the correct response of the child, the therapist will provide a reinforcement to the child in the form of a welcome reward (game, food, drink, etc.). In the case of an incorrect answer, the therapist will provide a form of correction that is adequate and useful relative to the child’s characteristics. During the described phases, the therapist, in addition to maintaining a high level of motivation for what is happening and keeping the end goals of the intervention in sight, has to monitor and collect data (using both paper documentation and/or technological support) that are useful for subsequent analyses, together with the working group, of the individual session as well as the ABA treatment in its entirety and evolution. Finally, during the session, technologies (smartphones, tablets, laptops, etc.) can represent the child’s real interests and stimuli, and they can be used as devices for specific tasks and gratify the child after a correct response.

Regarding an incorrect answer, clarification is important. In cases where we want to increase the probability that the child does not make mistakes because, for example, the result is particularly challenging or because the child is very oppositional, we can adopt a specific procedure called error-less learning/teaching. Within this procedure, we propose extremely facilitated activities because we want to obtain a highly positive result. For example, we reduce the time period of the child’s commitment to a minimum and at the same time offer him significant help. In this way, we attempt to greatly reduce the possibility of the child’s incorrect response and thus avoid potential problematic behaviors [120]. Also, in this case, technologies can facilitate data collection and represent stimuli that can be used to provide specific and rewarding tasks for the child.

### 6.2. Exploitation of Existing Solutions

Currently, therapists may rely on the use of different apps that can be adopted to ease specific operations that are performed during an ABA training session. Aiming at providing a real contextual example, Table 1 lists some existing apps that are mapped onto the ABA activities considered in the defined reference use case. As highlighted, the therapist can be supported by different tools, but all require real-time interactions for tracking a child’s behavior. This operation is sometimes unfeasible and requires a highly experienced therapist who is able to simultaneously perform the needed therapy interventions and data collection.

### 6.3. Exploitation of the Proposed Service Platform

Focusing on the three ABA operations considered in the reference use case (prompting, rewarding, and tracking), the main exploited features and capabilities of the proposed platform can be synthesized: (i) providing highly customized interventions based on the child’s profile and (ii) the real-time transparent monitoring of multiple behavior indicators. As the session starts, a therapist equipped with a tablet accesses the platform (e.g., using the platform app specifically designed for the client interface); opens the child’s record, which contains all child profile data, including historical data regarding their behavioral progress; and selects the target skill for the training session. On the other side, the child is equipped with wearable sensors that continuously collect their behavior-related data, which are transmitted in real time to the ABA ICT service platform. The collected data are processed by the platform’s AI algorithms, together with the selected target for the training session and the child’s historical data, with the aim to provide the therapist with prompt reward scheduling for that session. Differently from the adoption of existing the app, the possibility of including real-time and historical child behavioral data is a key element for the efficient and effective personalization of interventions. On the other hand, analyzing behavioral tracking that the therapist performs during the session, the platform can automatically gather behavior-related data directly from the sensors without the need of a continuous interaction between the therapist and the system, and this can be limited to some specific events that may occur. In detail, while all tracking apps rely on the insertion of data performed by the therapist during the session, the proposed platform autonomously collect some of these data. Moreover, the platform, based on the results of the real-time processing of the data, can adapt to and provide prompt reward scheduling and provide the therapist with a sort of online intervention guideline that can be followed during the session.In fact, the system may identify preliminary signs of an imminent stressful condition of the child based on indicators that are not observable but that can be detected by specific sensors. It is worth highlighting that automatic data collection may not be limited to the training session; it can also be used to monitor the child’s behavior during his/her daily life, increasing the amount of data that are useful for the understanding of his/her behavior intervention progress.

## 7. ICT Challenges for ABA Enhancement

The exploitation of ICT capabilities for designing and implementing advanced solutions based on a service platform, which include sensors for behavior monitoring and an AI engine for data analysis to support the ABA model’s application, involves many challenges. This section aims to provide an overview of some of the most critical challenges.

*User acceptance:* Defining ICT solutions—which are oriented toward the improvement of the effectiveness and efficiency of the techniques and approaches adopted for enhancing the social inclusion of people with disabilities—is itself challenging. Focusing on children with intellectual disabilities and the application of ABA therapy, intensive interactions with expert therapists are required in all phases of design, development, and testing processes. Deep engagement, interdisciplinary, individuality, and practicality need to be taken into account in order to create a solution that is suitable for end-users. User acceptance is essential for the success of an ICT solution: the users are able to use the new tools; the application behaves in a way it is expected to; the new tools provide the users with the expected solutions, easing their activities.

*Continuous monitoring solution:* Implementing a real-time continuous monitoring system that is able to collect different data that are useful for the child’s behavioral analysis requires the adoption of multiple sensors that should be embedded in a unique wearable device (e.g., a smart watch, plaster, etc.) making data acquisition transparent to the child and therefore not impacting his/her movements. Moreover, an adequate level of security and privacy should be guaranteed, taking into account both the computational capabilities of these devices and their energy resources. Recent advances in nanotechnologies may be help the design of new non-invasive devices that satisfy the aforementioned general requirements. However, the identification of specific requirements—also in terms of the needed body signs that are monitored and their translation to biometrically measurable indicators—is to be performed by behavior experts, ABA therapists, and engineers that are jointly working together.

*Interdisciplinary approach for AI Implementation:* In order to create a dynamic adaptive system that is able to efficiently support the application of ABA therapy (e.g., helping therapists in their daily activities and providing an easy guide for the child’s family), an interdisciplinary approach should be followed. ABA is a highly structured model that is based on a systematic approach in which observable and measurable data that are associated with a behavior are analytically evaluated. This characteristic makes the AI paradigm particularly suitable for supporting the methodological application of ABA therapy. However, one of the main challenges is to provide AI algorithms with adequate datasets for the training phase. This is fundamental for the realization of a highly valuable automatic system that is able to guarantee the needed support in many different application contexts and in line with the child’s functioning. Some activities required for achieving this objective are as follows: defining the body indicators that will be monitored; identifying how these parameters are linked to specific child behavior, taking into account his/her characteristics; linking the collected data to the identification of an antecedent and also processing additional environmentally collected information; evaluating the time progression of the collected data to prevent or decrease the consequences of negative behavioral effect (modeling the behavior–problem connection).

*Cost effectiveness:* One of the main issues of ABA therapy is economic sustainability. The ABA approach requires highly intensive activities that need to be supervised by behavioral experts, and these range from the child’s first and subsequent evaluations to family training and from the child’s personalized ABA program definition and update to its supervised application. All these elements are essential for satisfying the requirements of education coherence and the intensity of treatment, which are the basis for achieving objectives in learning and communication or, more generally, for improving both personal and social behaviors, such as personal autonomy or social relations. Defining a service platform based on cost-effective solutions and using a service platform that is able to support the application of the ABA model may reduce the cost of therapy treatments by decreasing the effort of supervisors in some of their activities. In these contexts, a deep understanding of the potentialities of such a system and the definition of updated procedures (logistic, organizational, technical, etc.) integrated with these new tools are two of the main challenges that have to be faced in order to provide families with an affordable service that is in line with the social inclusion paradigm.

## 8. Discussion

As detailed in Section 2, social inclusion for children with neurodevelopmental disorders should tend toward modifying cultures, organizational forms of contexts, and relational modalities in order to be able to respond to requests for learning and participation for all. Promoting inclusive paths and processes for children with disabilities represents a systemic and generalized action, and this can include the choice of a school as a privileged location for carrying out comparisons, research, and studies. As stated by the Behavior Analyst Certification Board [121], ABA “is best known for its success in treating individuals with ASD and other developmental disabilities (e.g., Down syndrome and intellectual disabilities). Treatment in this area is effective across an individual’s lifespan (i.e., childhood, adolescence, adulthood). In young children with developmental disabilities such as ASD, the goal of intensive, comprehensive intervention is to improve cognitive, language, social, and self-help skills”. As stated by Colombi et al. [122] in the conclusions of the research conducted on the Early Start Denver Model [123], the global increase in the early identification of ASD involves a need for early intervention, calling for interventions that are effective not only in rigorous research contexts but that are also valid and feasible in community contexts in which there are less available resources compared to university centers. For a child with intellectual disability or ASD, encountering difficulties in peer relationships and contexts translates, first of all, into a loss of opportunities with respect to learning. To counter this problem, the ABA approach involves parents, family members, teachers, and educators in an extended team that shares methods, procedures, and techniques in an attempt to achieve adequate, clear, and shared objectives. All this obviously entails costs, but the positive results highlighted in numerous research studies should allow family members and the community to perceive that these costs can become investments for the future of that child and for the reality of those close to him. In this complex and stimulating path, the identified role of ICT (described in Section 4, Section 5 and Section 6) is of fundamental importance. ICT provides advanced solutions for the continuous and context-aware monitoring of behavior, enabling automatic and real-time behavioral assessments. ICT allows rationalizing the exchange of information, the data collected, and the functional analysis within individual ABA treatments, avoiding the risk of losing or altering information and data. ICT allows the real-time processing of data and information and can share the obtained results among the ABA community, including the child’s family. ICT enhances some phases of observation and data collection within the ABA approach by rationalizing them and, therefore, making them less burdensome. The use of ICT in different environmental contexts, such as the domestic context, could capture significant elements that are unlikely to manifest in structured or otherwise less personal environments. To conclude, a potential line linking the role of ICT, the ABA approach, community contexts, and social inclusion is represented by the Early Start Denver Model [124,125]. It is characterized by the following noteworthy characteristics and overarching themes:An early intervention program to be implemented in nursery schools and kindergartens and aimed toward acquiring key behaviors in a small group context;Teachers as therapists who collaborate with healthcare professionals and university professors/researchers;The play area as a privileged context that is simultaneously structured and natural;Close collaboration with family members to generalize learning;The peer group as a reinforcement during flexible and rewarding activities;ICT as a support used in sustainable interventions that turn into tangible results, particularly for children and their characteristics;Results that become the heritage of the enlarged community, which fuel research and provide hope that something is changing.

In this scenario, the real-time, context-aware, remote, and continuous monitoring features of ICT allows it to become an effective and low-cost solution for sustainable ABA interventions.

To conclude, it is worth pointing out some considerations regarding ethical issues. Although ethical aspects are out of the scope of this paper, we mention [126,127] as two significant studies, both for their completeness and for further analysis in future studies. Key ethical considerations in the ABA approach encompass obtaining informed consent; ensuring privacy and confidentiality; prioritizing client wellbeing; personalizing interventions; using the least restrictive alternatives; maintaining professional competence; fostering collaboration and communication; monitoring and evaluating treatment; promoting transparency and accountability; and ensuring social validity. ABA practitioners must obtain consent and provide comprehensive information; safeguard personal information; act in the client’s best interest; tailor interventions to individual needs; prioritize non-aversive strategies; maintain professional competence; engage in collaboration; monitor progress; uphold transparency and accountability; and ensure that interventions align with values and goals for meaningful improvements.

## 9. Conclusions

This paper provides an overview of the potential of ICT to provide innovative solutions for fostering the social inclusion of children with neurodevelopmental disorders via ABA. It also discusses the challenges and limitations of using ICT to support ABA in social inclusion efforts for children with neurodevelopmental disorders. The role of ICT is identified as crucial in the ABA approach. ICT enables continuous and context-aware monitoring, facilitating real-time behavior assessments and data analysis. It streamlines information exchange and functional analysis within individual ABA treatments, minimizing the risk of data loss or alteration. ICT also allows for real-time data processing and the sharing of results within the ABA community and with the child’s family. Moreover, ICT enhances observation and data collection phases, making them less burdensome and capturing significant elements that may not manifest in structured environments.

In conclusion, the Early Start Denver Model represents a potential link between ICT, the ABA approach, community contexts, and social inclusion. This model features early intervention programs implemented in nursery schools and kindergartens, with teachers collaborating with healthcare professionals and researchers. It emphasizes the play area as a structured yet natural context, close collaboration with family members, and peer group reinforcement during flexible and rewarding activities. ICT serves as a support for sustainable interventions, yielding tangible results for children with neurodevelopmental disorders. These outcomes contribute to research and inspire hope for positive change within the community. The real-time, context-aware, remote, and continuous monitoring features make ICT an effective and low-cost solution for sustainable ABA interventions.

## Figures and Tables

**Figure 1 sensors-23-06914-f001:**
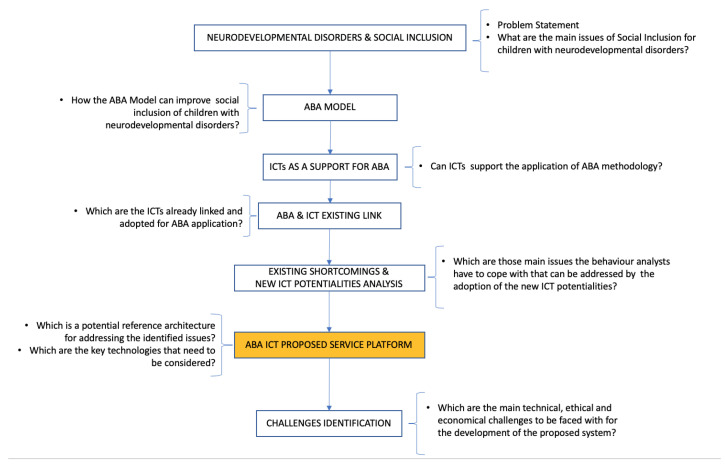
Research workflow.

**Figure 2 sensors-23-06914-f002:**
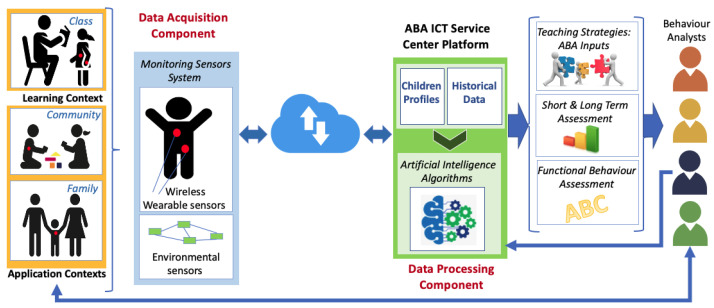
Reference to the ABA ICT service platform.

**Figure 3 sensors-23-06914-f003:**
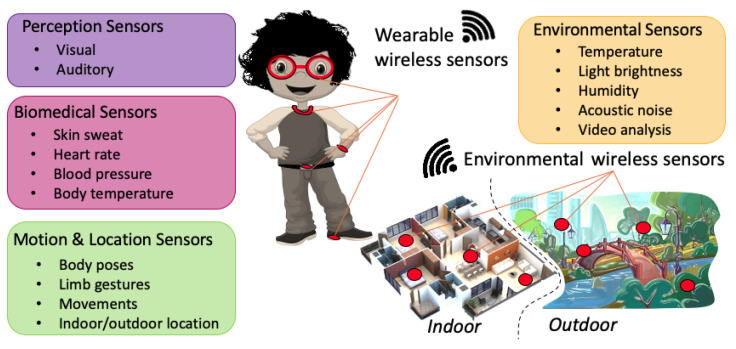
Wearable and environmental wireless sensors for ABA.

**Figure 4 sensors-23-06914-f004:**
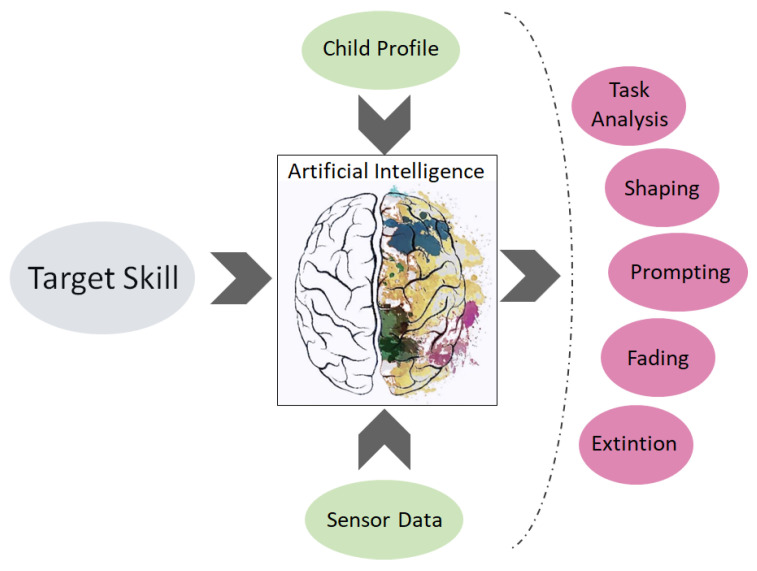
Applied behavior analysis enhanced by AI.

**Figure 5 sensors-23-06914-f005:**
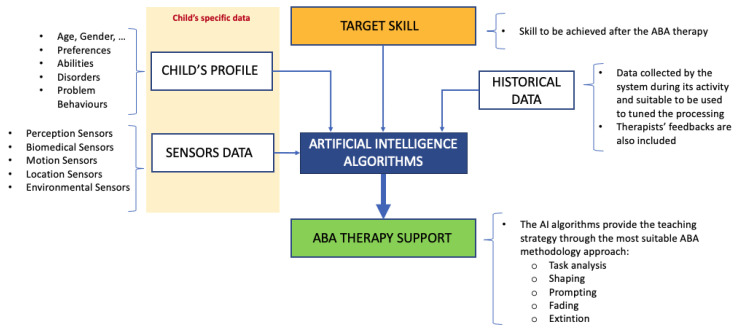
Representation of how AI algorithms can support the applied behavior analysis methodology.

**Figure 6 sensors-23-06914-f006:**
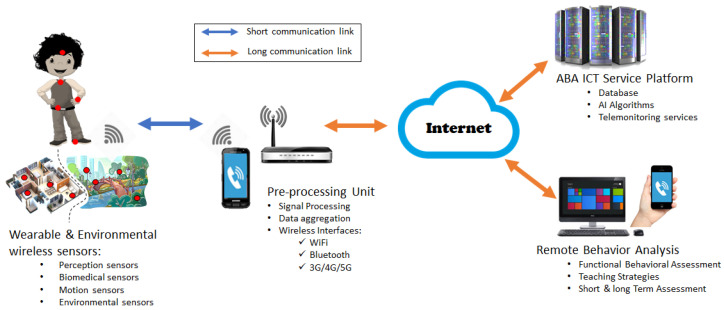
ICT Architecture for the ABA service.

**Table 1 sensors-23-06914-t001:** Existing ABA app for therapist that can be useful in the defined reference use case.

ABA Operation	App	Main Features
Prompting	iPrompts	- Picture-based scheduling - It provides image prompts - Countdown timer
Rewarding	iReward	- Target behavior selection - Reward selection - Reward policy definition - Track behavioral benchmarks
Tracking	Behavior Tracker Pro	- Behavior tracking - Intervention tracking - Video Recording - Data upload and sharing
	Skill Tracker Pro	- Target selection - Behavior progress tracking - Video capture - Organizational charts - Reporting
	ABC Data Pro	- Data-collection app - Goal setting - Behavioral tracking - Recording

## Data Availability

Not applicable.

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
