# Peer review of "ICT Framework for Supporting Applied Behavior Analysis in the Social Inclusion of Children with Neurodevelopmental Disorders"

_sensors, 2023, doi:10.3390/s23156914_

Round 1
Reviewer 1 Report
This paper investigates the use of technology in teaching children with neurodevelopmental disorders and proposes an ICT Framework for supporting and enhancing an ABA intervention, highlighting open challenges. Specifically, the potential of AI in the field of structured education and behavioral analysis is explored for monitoring and improve the effectiveness of the intervention over time.
The paper is simple to read, well-organized and logically structured. The research questions are nicely shown in a flow diagram.
Section 4.2. Background can be enriched including previous experiences of ABA technology-enhanced intervention in children with autism such as:
Artoni, S., Bastiani, L., Buzzi, M. C., Buzzi, M., Curzio, O., Pelagatti, S., & Senette, C. (2018). Technology-enhanced ABA intervention in children with autism: a pilot study. Universal Access in the Information Society, 17, 191-210.
Parsons, D., Cordier, R., Lee, H., Falkmer, T., & Vaz, S. (2019). A randomised controlled trial of an information communication technology delivered intervention for children with autism spectrum disorder living in regional Australia. Journal of Autism and Developmental Disorders, 49, 569-581.
Line 759. Authors affirm “In case of an incorrect answer, the therapist will provide a form of correction, adequate and useful to the child’s characteristics.”
Authors have to consider the error-less principle of the ABA intervention. Errors are very difficult to be corrected thus therapists must avoid the child making an error... for instance, in a discriminative program, therapists must avoid the child making a wrong match, by stopping his/her arm, saying no, and bringing the piece in the original position. Then they can provide a prompt to avoid the error. Analogously the software has to respect the error-less principle by implementing a repulsive movement that brings the object to the original position in the screen in case of a possible error (impeding the vision of a wrong match) (see Artoni et al. 2018).
In the discussion ethical considerations are missing.
Abstract and Conclusion might be a little bit more focused on the work.
Reviewer 2 Report
Dear authors,
This is an interesting article for the scientific community where the relevance and importance of ICT for the support of children with neurodevelopmental disorders is evidenced with the support of a multitude of previous studies.
I am very concerned when I read the article, as there are currently many health and social professionals who consistently reject the use of the ABA method, as well as scientists who have addressed the issue. In countries like Spain or other South American countries, this way of working is totally rejected in the therapy of children with autism or other neurodevelopmental disorders.
However, I propose some improvements as for the content of the work carried out:
1. Summary: we start describing the aim of the work. This is a mistake, as it is advisable to start with a brief description of the subject. It is suggested to improve this aspect. It is also recommended to put the meaning of the acronym ICT, just as it is done with ABA.
2. The objective of the research is presented in a rushed manner in the introduction. This should be proposed later on.
3. It is recommended to put "information and communication technologies" (lines 21-22) and "applied behaviour analysis" (line 25) with initial capital letters.
4. It is recommended to always use ITCs instead of ITCs. The "s" is not usually included. In fact, authors sometimes use ITCs and sometimes ITCs.
5. It is recommended to update the DSM-5 reference to the revised and updated version in 2022 (DSM-5-TR).
6. The acronym ASD is described on line 137, but is not used again in subsequent paragraphs. For example, in line 321. This should be changed throughout the document and ASD should be used in all cases for smoother reading.
7. In lines 179-18, the page number from which the quotation is taken should be given, as it is a verbatim quotation.
8. In line 322 all the numbers can be put in the same bracket, for example: [30-35]. This should be changed throughout the work.
9. The figures are appropriate and allow a better understanding of what is explained in the text.
10. References do not follow MDPI citation guidelines. They should be improved considerably.
11. Only 20 of the references used in the entire work (113 references) are current and belong to research carried out in the last five years. This is unacceptable in the current situation, as the research on ICT for learners with ASD is extremely high. It is strongly recommended that the sources used be updated.

Round 2
Reviewer 2 Report
Dear authors,
The suggested changes have considerably improved the article.